# Intelligent Healthcare: Integration of Emerging Technologies and Internet of Things for Humanity

**DOI:** 10.3390/s23094200

**Published:** 2023-04-22

**Authors:** Van Anh Dang, Quy Vu Khanh, Van-Hau Nguyen, Tien Nguyen, Dinh C. Nguyen

**Affiliations:** 1Department of Information Technology, Hung Yen University of Technology and Education, Hungyen 160000, Hungyen, Vietnam; dangvananh@utehy.edu.vn (V.A.D.); haunv@utehy.edu.vn (V.-H.N.); 2Department of Electrical and Electronics Engineering, Lac Hong University, Bien Hoa 810000, Dong Nai, Vietnam; chitien2802@gmail.com; 3Department of Electrical and Computer Engineering, University of Alabama in Huntsville, Huntsville, AL 35899, USA

**Keywords:** smart healthcare, 6G, Internet of Things, edge computing, fog computing, cloud computing

## Abstract

Health is gold, and good health is a matter of survival for humanity. The development of the healthcare industry aligns with the development of humans throughout history. Nowadays, along with the strong growth of science and technology, the medical domain in general and the healthcare industry have achieved many breakthroughs, such as remote medical examination and treatment applications, pandemic prediction, and remote patient health monitoring. The advent of 5th generation communication networks in the early 2020s led to the Internet of Things concept. Moreover, the 6th generation communication networks (so-called 6G) expected to launch in 2030 will be the next revolution of the IoT era, and will include autonomous IoT systems and form a series of endogenous intelligent applications that serve humanity. One of the domains that receives the most attention is smart healthcare. In this study, we conduct a comprehensive survey of IoT-based technologies and solutions in the medical field. Then, we propose an all-in-one computing architecture for real-time IoHT applications and present possible solutions to achieving the proposed architecture. Finally, we discuss challenges, open issues, and future research directions. We hope that the results of this study will serve as essential guidelines for further research in the human healthcare domain.

## 1. Introduction

The history of human development has proven that healthcare applications have always been the main driving force behind the development of science and technology. Since ancient times, humanity has envisioned solutions to diagnosing and treating diseases from a distance. The advent of $5^{\mathrm{th}}$ generation network technology, also known as 5G, enables the delivery of network services with ultra-high throughput and ultra-low latency. This has led to the Internet of Things concept and shaped emerging intelligent domains. Among these areas, healthcare and medicine are becoming some of the most crucial domains.

In recent decades, electronic health information systems (e-health) have focused on developing and achieving many positive results. In the traditional architecture of e-health systems, computing, processing, and storage are located in the cloud. Its robustness, reliability, and powerful computing capacity make cloud computing (CC) the computing technology of the future [1,2]. Although cloud computing has outstanding advantages, a significant limitation of CC is its high service response time. As a result, it is impossible to use in real-time healthcare applications. In recent times, to solve this problem, the integration of IoT technology and fog computing (FC) solutions [3] with edge computing (EC) [4] has been proposed.

EC and FC technologies aim to bridge the gap between databases and end-users or bring cloud capabilities closer to users. As a result, FC and EC reduce service response times, energy consumption, and computational costs, and improve reliability.

According to Statista’s forecast, the IoT market is expected to reach USD 75 billion by 2025 [5]. Moreover, a survey by Vodafone shows that over 77% of people surveyed spend more on intelligent health. In addition, about 60% of people surveyed use IoT devices to monitor vital indicators such as blood pressure, heart rate, and blood sugar [6]. These issues show that healthcare focuses on IoT technology and real-time computing solutions.

Telemedicine is one of the most significant medical applications in recent times based on IoT technology. Telemedicine can bridge the gap between rural and urban health. It provides low-cost consultation, remote examination, and diagnosis. As a result, Telemedicine can enable leading experts and doctors to reach the most remote areas and provide advanced medical services to everyone at a low cost. In other words, combining IoT technology and real-time computing solutions plays an essential role in the healthcare sector.

There are many studies on the IoT, and advanced computing technologies have been proposed in the field of smart health. To clarify this matter, we conduct a short survey of studies that have been conducted during the period 2019–2022 on the aspects of computing methods, vision, architecture, challenges, and emerging technologies.

We evaluated these aspects at three levels (High (H), Medium (M), and Low (L)), depending on the levels at which they are mentioned in each study. Thus, we searched for articles using keywords such as Intelligent Healthcare/Medicine, Smart Healthcare/Medicine, Computing Technologies, and Emerging Technologies in electronic databases such as IEEE Xplore, ACM, ScienceDirect, Springer, and MDPI. Then, we evaluated the search results using three-step processing. In the first step, the searching step, we identified 450 related works. In the second step, we removed duplicate and inappropriate papers, and the remaining number of articles was 220. In the final step, we discarded papers from low-reputation journals and conferences, resulting in 95 papers, as presented in the References section. 

Based on our survey results, we evaluated the studies based on criteria such as vision, architecture, challenges, and technology solutions, and outlined the key findings of each study. The details of the survey results are presented in Table 1.

The survey results show that smart healthcare systems are receiving substantial attention from academia and industry. Smart healthcare applications use one of three computing solutions (CC [7,8,10,16], FC [12,14], or EC [21]) or a combination of solutions [9,11,14,15,17,20].

Some studies also point out visions [8,16], propose improved architectures [7,8,11,12,14,15,20,21], and point out the challenges in this domain [11,19,20].

One of the most important aspects is that of technological solutions. The survey results show that many possible technologies are integrated to enhance the capabilities of smart healthcare systems, including the IoT [10,19], AI [11,21], WBAN [8], and blockchain [11,16]. Androcec [22] reported, in a recent survey, on the application of IoT solutions to monitor the COVID-19 pandemic. The author found over 155 related works in this area. The survey results indicated that IoT technology can be applied to almost all stages, such as contact tracing, health monitoring, social distancing, diagnostics, and treatment.

Although some studies have integrated one or several technologies to improve the performance, energy efficiency, or privacy and security of smart healthcare systems, the latest survey results have shown that real-time responsiveness, privacy, and security remain significant challenges for smart healthcare systems. 

In this study, we conduct a comprehensive survey of IoT-based medical and healthcare applications. We propose an all-in-one computing architecture for real-time healthcare applications and indicate application directions for the proposed architecture. Finally, we discuss some challenging aspects, open issues, and future research directions. The main contributions of this study are as follows:We comprehensively review several key Internet of Things-based medical and healthcare applications over the past three years to highlight possible research directions.We investigate the core technologies and emerging technologies that enable smart healthcare applications.We propose an all-in-one computing architecture to reduce ***service response time***, ***computation costs***, and ***energy consumption*** for real-time IoHT applications.We indicate application scenarios for the proposed architecture.Finally, we discuss challenges, open issues, and future research directions.

The rest of the paper is organized as follows: In Section 2, we consider core technologies for Internet of Health Things applications. Section 3 presents emerging breakout technologies for IoHT applications. In Section 4, we describe a full picture of the applications and use cases of IoT-based healthcare systems. We propose an all-in-one computing architecture for real-time smart healthcare systems in Section 5. Section 6 indicates challenges and some open issues. Finally, our conclusions are presented in Section 7.

## 2. Core Technologies for Smart IoHT Applications

Some advanced technologies and solutions have been researched, developed, and deployed in the smart healthcare field, such as smart sensors, autonomous devices, robots, intelligent computing solutions, and virtual reality. In this section, we present some advanced core technologies.

### 2.1. Smart Sensors

A smart sensor is a device that allows users to accurately and automatically collect data on physical and chemical changes in the area where the device is mounted. Intelligent sensors collect information with high accuracy.

In this study, we focus on wearable sensors because of their universal applicability in IoHT applications. Wearable sensors can be embedded in clothing, implanted in the body, or worn on the wrist. In smart healthcare, devices collect patients’ vitals and transmit them to calculation servers via wired or wireless network connections. Many wearable products have been focused on development in recent years [23], such as smartwatches, wristbands, eyewear, headphones, earplugs, body straps, and devices worn on the hands and feet, as presented in Figure 1. In [24,25], the authors indicate feasible solutions to the application of wearable medical sensors by using varying smartphone sensors to detect anomaly data in healthcare areas.

Sensors can be classified as wearable sensors, implantable sensors, etc. In the smart healthcare domain, wearables can be mainly used to measure and check patients’ activities and vitals to alert or provide data to healthcare systems from a distance. Some of the patient vitals that need to be monitored based on wearable sensors are as follows [26]. 

Pulse: A pulse sensor monitors the pulse in the human body, and can be used to monitor emergency conditions such as cardiac arrest and pulmonary embolism. The pulse signal can be installed in positions such as the wrist, earlobe, chest, fingertip, etc. Signals from the earlobe and fingertip locations are highly accurate, but placing the pulse sensors in these areas is difficult. Sensors placed in the wrist position are often more convenient and long-lasting.

Respiratory Rate: A respiratory rate sensor measures the respiratory rate, or the number of breaths of a patient per minute; it is used to monitor individuals with airway-related diseases such as asthma, lung cancer, respiratory failure, tuberculosis, etc. 

Body Temperature: A body temperature sensor is used to measure a patient’s temperature. These sensors can read the appropriate temperature range to monitor the temperature of the human body. The accuracy of the body temperature sensor depends on the location of the human body.

Blood Pressure: A blood pressure sensor is used to measure a patient’s blood pressure (BP). High blood pressure is a risk factor for cardiovascular disease. However, the accuracy of a blood pressure sensor depends on its measurement location. Since there is no precise BP measurement, several modalities have been proposed whereby two PPG optical sensors are located at different positions on the patient’s arm.

Oxygen: An oxygen sensor measures the oxygen level in the blood. This is an important parameter to assist doctors in accurately diagnosing the amount of oxygen supplied to the body. However, due to the use of infrared LEDs, a major limitation of these sensors is their high power consumption. To solve this problem, measurement techniques based on the signal-to-noise ratio (SNR) and PLL [27] were proposed. The results showed that using this solution can save six times more energy than not using this solution. 

Finally, the data collected from the sensors is transmitted to dedicated servers in different computing layers for processing, calculation, and storage to provide optimal intelligent medical services to patients. In the next section, we will discuss some of the advanced computing technologies in the IoHT domain.

### 2.2. Cloud, Fog, and Edge Computing

Cloud computing (CC) has existed for decades. One of the key unique characteristics that make CC successful is its ability to provide everything as a service, including software, infrastructure, and platforms. Over the years, power and flexibility have made CC the dominant computing technology for information and communication systems. The basic CC model consists of two layers: the cloud and end-users. The cloud layer includes servers with powerful configurations, high computing power, and large storage capacity. Cloud servers are connected to the Internet infrastructure via back bolt connections with extremely high throughput. The end-user layer includes end-users such as sensors, IoT devices, actuators, etc. These devices are connected to the cloud layer based on wired and wireless connections. Processing, computation, and storage all take place on cloud servers. One major limitation of CC is its high service response time. Therefore, it is not viable for emergency healthcare scenarios that require real-time computing power.

In recent years, to solve this problem, several computing models, such as fog computing (FC) [28] and edge computing (EC) [29], have been proposed. Figure 2 presents an all-in-one computing framework for IoT systems.

Both FC and EC aim to bring database and cloud capabilities closer to the end devices. One key difference between EC and FC is their computational locations. While EC computation is integrated into devices, the edge server is set up in LAN, a few hops away from end devices; in contrast, FC computation is performed in the dedicated servers of data centers, set up from the Internet gateway to the cloud.

Each computing model has its advantages and disadvantages. CC has high computing power, but a considerable delay. EC and FC have low service response time, but inadequate computing and storage capacity. However, for each specific scenario and application, we can combine these technologies into an optimally integrated solution. To explain the capabilities of computing technologies, performance comparison studies are presented in Table 2.

To collect patients’ vitals, each patient can carry a variety of sensors, depending on the type of disease and the indicators to be monitored. These sensors assist each other in data acquisition, communication, and the formation of Wireless Body Area Networks (WBANs). The issues with WBAN will be discussed in detail in the next section.

### 2.3. Wireless Body Area Networks (WBANs)

In smart healthcare, a WBAN is one of the most important core technologies. A WBAN is a set of intelligent IoT devices mounted on the human body to collect patient vitals and transmit this information to a data center for decision making. A WBAN works via wireless technology consisting of light and smart sensors attached to or implanted in the human body, measuring parameters such as heart rate, body temperature, blood glucose level, etc. These data are then visually provided to medical staff or caregivers to monitor and analyze the patient’s condition and devise appropriate treatment schemes [30]. 

The WBAN architecture consists of hardware, software, and communication technology. The hardware includes all devices and sensors used to collect patient data; the software consists of a man–machine interface, network protocols, and an operating system. In addition, communication technology in a WBAN plays a significant role in transmitting medical data between devices and software. Figure 3 illustrates the architecture of a typical WBAN system. Depending on the goals of each WBAN network, different communication technologies will be used. Table 3 compares the characteristics of some communication technologies of WBAN systems [31].

In recent years, to respond to more increasing requirements of humans in the smart healthcare domain. Besides existing core medical technologies, some emerging technologies are being researched and are expected to provide booming results, including the Metaverse, digital twins, and AI. In the next section, we will discuss detail these issues.

## 3. Emerging Technologies in Smart Healthcare

The COVID-19 pandemic has exposed the limitations of existing healthcare systems and created the concept of social distancing for the first time. These issues require a disruptive change in the smart healthcare sector. Therefore, AI, digital twins, and Metaverse technologies answer these demands. Recently, these technologies have received special attention from the research community and undergone several feasibility studies, which are summarized as follows.

### 3.1. Metaverse

The definition Metaverse was created by writer Neal Stephenson in his science fiction novel Snow Crash in 1992 [32]. In this novel, Stephenson describes the Metaverse as a vast virtual environment that coexists with the physical world, in which humans interact through digital avatars. A Metaverse is a shared three-dimensional space in which users can perform all actions through virtual reality technologies such as AR, MR, and VR [33], as presented in Figure 4.

Nowadays, the Metaverse is described as the future of the Internet. In [34], the authors showed the vision of Metaverse developments in the medical domain, including telemedicine, clinical treatment, medical training, mental health, fitness, medicine, and pharmacy. Despite its advantages, a series of challenges must be addressed before the Metaverse can be widely adopted. We summarize the key challenges of Metaverse technology in Table 4.

Hardware: The Metaverse relies on virtual reality technologies. These technologies require the processing of a huge amount of data with high computing costs and real-time service response. Hence, the hardware should be designed to be smart, flexible, and smaller at a more affordable cost. These will be significant challenges for the popular Metaverse [35].

Privacy and Security: Several studies have shown that the collection and processing of data from users can compromise their data privacy and security [36,37]. In our opinion, the guarantee of privacy and security is one of the most crucial conditions in realizing the Metaverse, especially in the healthcare domain. If privacy and security issues are unguaranteed, they could lead to real-world disasters. 

Identity Hacking: In the Metaverse, clones can be hijacked (i.e., identity theft), and then, the hijackers can perform illegal actions, spread fake information, or steal users’ identities. Consequently, actions taken in the virtual world can have real consequences for humans in the real world [38].

Neurology-Related Diseases: The Metaverse allows humans to interact with their friends through their avatars, and attend virtual events. Consequently, humans are becoming further and further away from real life, which can lead to health problems and neurological diseases [35,39].

Digital Currencies and Payments: The Metaverse is more than an entertainment platform. It will be a global online marketplace with billions of users, where users can use currencies and cryptocurrencies to make payments or perform fast and smooth transactions. However, ensuring the security of e-commerce transactions will be a significant challenge [35].

Law and Policies: The development of the Metaverse will require the formation of new legal and policy concepts with the concepts of virtual citizens, virtual crimes, and virtual and flat worlds. Blocking a user account will not be enough to prevent illegal acts. Instead, new legal policies need to be announced to manage the Metaverse world [35].

Despite the dozens of challenges presented above, in our opinion, it is inevitable that the Metaverse will become a trend in development. Nowadays, research on the Metaverse is only at a primitive level, so we will need much breakthrough research in the future to realize the Metaverse concept in the medical domain, as well as in public life.

### 3.2. Digital Twins

Digital Twin technology is a combination of virtual reality technology, big data processing, and 3D graphics to build virtual models of processes, systems, services, products, or physical objects [40]. Through digital twin technology, users can experience virtual effects, identify problems before they happen, or predict future outcomes.

In the healthcare domain, digital twin technology is considered to be revolutionary and is applied in the prediction and early diagnosis of diseases that may occur in patients by examining organs or symptoms in the body. In the case of atherosclerosis, it can be used to perform vascular surgery to restore blood flow, replace the lenses of cataract patients, transplant organs, etc. [41]. A patient’s digital twin is created as a result of transferring the physical characteristics and changes in the patient’s body to a digital environment. This technology enables accurate diagnosis and delivers tailored treatment protocols in real time to patients. Figure 5 presents an illustration of digital twins in the healthcare domain.

In our opinion, digital twins can be extensively applied in the smart healthcare domain in the future. For example, before physical interventions such as drug treatment, radiation therapy, or surgical operations, paramedics can perform processes on virtual digital twins to determine the optimal treatment for the patient. By creating a digital twin of the patient’s body, doctors can perform a diagnosis by examining the vital areas of the patient’s digital twin without a direct impact on the patient’s body. With the creation of digital twins of medical devices, the prediction and coordination of medical resources for medical examination and treatment have high accuracy and optimization of system resources.

### 3.3. Artificial Intelligence

Smart healthcare requires the use of a large number of IoT devices and smart sensors to collect data continuously from patients. Handling these huge amounts of data requires advanced data processing technologies. AI technology is the answer to these problems [42].

In reality, AI has existed for decades. AI is the ability to equip machines with human intelligence. Today, AI is applied in almost all intelligent applications that serve humans, from cleaning robots [43] to military weapons systems [44]. In the smart healthcare domain, AI contributes to all stages, including early diagnosis, disease identification, and treatment [45,46]. To suit different intelligent applications, a series of improved AI techniques have been proposed, including ML, DL [47], RL [48], and DRL [49]. 

It is notable that that traditional AI techniques use centralized data management, whereby data are centrally managed on servers for training. The aggregator then uses these data to train and create optimal models. However, in smart healthcare, due to the sensitivity of medical data, ensuring the privacy of patient data is one of the issues of particular concern. Therefore, it is impossible to manage patient data for AI training centrally.

In our opinion, distributed learning techniques should be considered; training should be implemented on patients’ IoT devices. This will not cause privacy issues regarding medical data; moreover, it will not put pressure on backbone links by transferring all patient data to training servers.

The purpose of AI in healthcare is to determine the relationship between patients’ information and a fitting treatment approach [50,51]. Varying AI techniques have been applied to different disease scenarios. In [52], the authors propose an AI-based recognition and image diagnostic solution. In [53], the authors propose an AI-based remote sensing image retrieval algorithm by improving the Sobel method. In [54], the authors propose an AI-based data dimension-reducing algorithm to handle big data applications that have large-scale and high-dimensional to perform criminal detection or solve smart city problems. In [55,56], the authors propose DL technique-based image diagnosis algorithms for handling fault images.

In the drug medicine domain, patient monitoring supports doctors in their decision making regarding the personalization of treatment and prescriptions [57]. Moreover, AI-based systems also support doctors in seeking related medical information from online libraries, magazines, and textbooks [58], and in the storage of medical data via cloud solutions for convenient access. In [59], the authors present a comprehensive survey of FL-based IoHT applications where patients will obtain complete medical support throughout their lives. The authors also indicate that AI can be combined with most healthcare units, including emergency medical units, medical staff, diagnostics, laboratories, and pharmacies.

Figure 6 shows that AI can be integrated with most healthcare systems to provide optimal solutions. AI can assist a patient from the moment they are admitted to the hospital through integration into emergency medical applications, process patient data, detect serious illnesses, automatically identify complex samples, and analyze complete human and patient molecular data in a clinical setting. AI can assist doctors and medical staff through highly accurate clinical reports and provide many other decision-support tools. AI effectively supports optimal treatment decision making for patients. Recent studies demonstrate that AI strongly assists in detecting cancer at an early stage.

### 3.4. Blockchain

In the Internet of Things era, smart devices, sensors, and IoT applications are connected to others based on the Internet infrastructure. This leads to real challenges regarding IoT applications in terms of privacy and security. Indeed, attackers can illegally access and hijack the system or steal data through the security vulnerabilities of IoT devices [60], operating systems [61], Internet gateways [62], remote computing servers [63], or denial of service (DDOS) attacks [64]. In the smart IoHT domain, medical data records play a particularly crucial role and are sensitive [65]. 

In reality, blockchain technology is an advanced database management mechanism that enables transparent information sharing across a distributed network environment. Data are stored in blocks that are linked together in a consistent chronological sequence. Hence, the user cannot delete or modify the chain without the consensus of all users. As a result, blockchain technology creates an immutable ledger that keeps track of data records [66]. In the smart IoHT domain, blockchain technology is applied to ensure privacy and security in managing patient medical records. In recent years, integrating blockchain into healthcare applications has attracted strong interest from the academic community and achieved some positive results. In [67], to enhance privacy and security, the authors integrated blockchain into MEC-based IoT applications. To solve the delay caused by blockchain, they used a lightweight block verification algorithm. To ensure a suitable edge environment, they also applied a novel DRL-based AI technique. Their results showed that the proposed scheme significantly improved performance, privacy, and security compared to existing solutions. In [68], the authors designed a novel secure mobile edge computing framework for healthcare systems, namely BEdgeHealth. Indeed, they integrated blockchain into MEC-IoHT systems to enhance privacy and security for health data record-sharing and improve QoS. The real-world evaluation results showed that compared to existing computing solutions, the BEdgeHealth framework robustly improved QoS, privacy, and security.

In our opinion, IoHT applications will be impossible if privacy and security are not guaranteed, and blockchain technology is the key to solving this problem. Figure 7 presents an illustration of Blockchain-based smart health systems.

## 4. IoT-Based Smart Healthcare Systems

In this section, we present IoT-based smart healthcare systems. Based on our survey results, we divided these applications into four different approaches based on the objectives of the studies, as modeled in Figure 8 and statistically in Table 5.

### 4.1. Real-Time Monitoring and Alarm Generation

Monitoring health metrics such as temperature, heart rate, and blood oxygen is critical to delivering real-time healthcare services. Through the IoT, sensors are attached to the human body and measure various indicators, which are then analyzed to recommend drugs for the emergency treatment of patients. Shain et al. [69] developed an electronic medical system that monitors electrocardiograms, temperature, foot pressure, and heart rate. This system determines the patient’s GPS location to provide urgent care, and uses RFID to identify each patient, the Arduino Uno mainboard as the microcontroller, and Thingspeak as the middleware for medical signal handling. Swaroop et al. [70] proposed a real-time IoHT system that relies on the Raspberry Pi 3 platform and some sensor types, such as DS18B20 and Sunroom sensors, to monitor blood pressure or heart rate. Ratho et al. [71] designed a real-time IoHT application that relies on Apache Spark and the Hadoop platforms to handle big data and aims to reduce response time. Their experimental results demonstrated the effectiveness of the proposed solution in handling the big data of smart cities or countries.

### 4.2. Telemedicine

Telemedicine allows for the provision of remote medical services and relies upon Internet infrastructure and communication technologies. Overall, these enhance the response capacities of medical services and staff, improve patients’ health, and reduce treatment costs. In [72], Zouka et al. introduced an AI-based IoHT system to analyze collected data from medical devices mounted on patients’ bodies. The information collected from medical sensors is transferred through the GSM system to the Azure data center for data handling and decision making. In [73], Rohokale et al. designed a novel IoHT system for rural residents by monitoring their main survival indicators. The patients wore an RFID tag for identification.

When patients’ vital indicators, such as blood pressure and heart rate, have abnormal changes, the IoHT system will generate alarms, and then, send them to doctors, hospitals, or caregivers. Mohammed et al. [74] designed an IoHT system for remote patient monitoring by combining web services and cloud computing-based solutions. Indeed, the authors developed an ECG Android App to monitor the electrocardiogram indicators of patients. The obtained signals are transferred to Microsoft’s Azure platform for handling. Moreover, they also used a hybrid cloud, where patients’ sensitive medical records are pushed into private clouds, and general medical information is pushed into public clouds.

### 4.3. Chronic Disease Detection and Prevention

A massive number of patients face serious diseases such as heart disease, diabetes, cancer, etc., which then cause depression in patients. To solve this problem, Sundhara Kumar et al. [75] designed an IoHT system to monitor autistic patients. The application measures and collects EEG waveform signals through neural sensors, and then, alerts are sent to caregivers in case any abnormal results are detected. Onasanya et al. [76] have proposed different architectures and frameworks that support IoT-based healthcare solutions for cancer patients. The focus is on cloud services, which use big data technology to analyze data over the air. Sood et al. [77] designed an IoHT system to monitor the *chikungunya* virus pandemic. The system uses medical sensors to collect medical information, and then, the system transfers this information to a private cloud. From there, fog computing methods combine fuzzy logic systems and aim to detect the possibility of infection in patients and immediately alert hospitals and caregivers. Sensitive medical records related to the patients’ information is pushed into the private cloud for privacy and security.

### 4.4. Home Healthcare and Healthcare for the Elderly

IoT and reality technologies can be deployed at home to continuously monitor elderly people who move slowly and take longer to arrive at the hospital for routine or urgent health care services. Abdelgawad et al. [78] proposed a life support health monitoring system. Indeed, the authors used varying sensor types to collect medical signals, and then, the information was transferred to the cloud infrastructure for handling and big data analyses. Additionally, they also designed a prototype to prove the effectiveness of the proposed solution. This system consists of six medical sensor types: a Bluetooth-based communication module, a Raspberry Pi-based microprocessor, and a Wi-Fi-based communication module. All the medical signals are stored in cloud infrastructure.

Yang et al. [79] proposed a system of individualized health care for people who live alone and use wheelchairs. The system combines IoT technology and WBAN technology to provide efficient medical solutions for wheelchair patients by monitoring real-time heart rate, ECGs, blood pressure, and environmental indicators. Cerina et al. [80] presented a new method for patient health monitoring using IoT technologies. Patients are fitted with medical sensors to monitor ECGs and survival indicators, and then, this medical signal is transferred to cloud servers that rely upon wireless links.

The IoHT systems analyzed above have varying requirements. Some IoHT systems have no service response time. Contrarily, others have strict requirements for service response time. The computing architectures directly affect the system performance. In the next section, we will discuss several recent existing computing architectures for IoHT systems.

## 5. Proposed Architecture for IoT Healthcare Application 

Some conditions are significant and need to be treated promptly, such as cardiovascular diseases or conditions that occur in patients following dangerous accidents that affect their lives. Such situations require fast real-time action with minimal delay. In a general cloud environment, data are transmitted to the cloud, processed in the cloud, and receive a response, which takes a long time and involves considerable delays. To overcome or limit latency issues, we can use fog computing, which brings computing devices and storage resources closer to the edge of the network. Most current healthcare solutions use the cloud’s decision-making environment. In recent years, many other proposed solutions have considered fog computing for healthcare applications where the time factor is of interest. Some architectures for existing solutions are shown in Table 6.

According to our survey results, traditional three-layer architectures are presented for IoHT applications in [20,83,85,86,87,88,89,90], including the sensor, fog, and cloud layers. The use of techniques to increase the security of identity management, user authentication [86,88,90,91], or the application of emerging technologies in healthcare, such as AI, WBAN, big data, and blockchain [86], is recommended by the authors.

The authors of [81,91] use a four-layer architecture based on FC to support healthcare, namely the physical, boundary, fog, and cloud layers.

Verma et al. [82] proposed a five-layer FC architecture for remote patient health monitoring that includes data acquisition, event classification, information mining, decision making, and cloud storage layers. Another study also proposed a five-layer computing architecture that includes sensing, transport, processing, application, and business layers [14]. Our survey results show that integrating various computing solutions with CC reduces service response time and realizes real-time smart healthcare applications. However, one limitation of the existing proposals is that there needs to be a fully integrated architecture framework for all existing solutions in the smart healthcare domain. 

In this study, we propose a fully integrated architectural framework for computing solutions for IoT-based healthcare applications to optimize service response time, compute costs, and realize real-time smart healthcare applications, as shown in Figure 9.

The proposed architecture includes four layers (the things layer, edge computing, fog computing, and cloud computing), where the things layer includes smart IoT devices, health sensors, actors, ambulances, etc. The main task of this class is to collect medical data and patient vitals in real time. The collected data are sent to local server nodes based on wired or wireless connections and communication technologies for EC.

The edge nodes are deployed in the LANs of hospitals or patient treatment places. As a result, computation is performed closest to the end-users, minimizing transmission delays, reducing data load on backbone connections, and improving computing performance. Depending on the processing results, the results can be sent directly to doctors, families, and emergency services (notification module) or sent to higher calculation layers for further processing.

The FC layer is operated by computing servers that are deployed at cloud gateways or computing service centers, also known as cloudlets. The data sent up from the lower layers are analyzed, processed, and aggregated via FC. Although FC’s transmission delay is higher than EC’s, FC’s storage and computing capacity are better than EC’s. As a result, FC can handle more complex problems than EC. Like EC, FC’s results can be sent directly to the message module or sent to the CC for further processing.

Instead of performing computations like traditional solutions, the CC layer in this architecture will receive the results from the FC layer and analyze and store the data to perform big data computation tasks, such as statistics and disease diagnosis, in the medical domain.

A comparison with existing computing architectures is presented in Table 6. Our architecture has many outstanding advantages. The parameters were considered to compare the different architectures of the layers, the architectures’ complexity, the data’s reliability in the fog layer, the real-time application support, and the security. The values for the selected parameters can be any of the following: *low*, *moderate*, or *high*. The value for complexity was chosen based on the classes and the function or module implemented in each class. The value for the reliability of the data in the fog layer was chosen based on the availability of clusters or distributed computers in the fog layer. No existing architecture emphasizes data reliability in the fog layer, which is a major concern for urgent care applications. The value for the real-time support application was selected based on the existence of the fog layer and the amount of work performed in this layer. Finally, security issues in the layers were taken care of. Most of the previous jobs should have emphasized the security of the architecture.

The use of the IoT as a technology in healthcare is still in its infancy. Therefore, there are still many challenges that need to be addressed by the research community and industry. Some of the existing challenges and solutions are discussed in the next section.

## 6. Challenges and Open Issues

Nowadays, smart healthcare systems are growing explosively, in both number and scale, due to increasing human needs. Despite the the positively achieved results, smart healthcare systems face several challenges [92]. In this section, we discuss some important challenges, such as issues with fault tolerance, latency, power efficiency, interoperability, and availability, as presented in Figure 10.

Fault Tolerance: The reliability of an IoT-based healthcare system is affected by the operation of the sensors and communication nodes that pass data onto the computational layers above. Reliability is one of the most important elements of a smart healthcare system, especially in emergency scenarios. In [93], the authors proposed a method of using redundant IoT nodes to improve the fault tolerance of the system.

Latency: The lag time of smart healthcare applications directly affects the quality of medical services. The main factors affecting delay time are transmission delay in the network layer and delay due to the computation and processing of services. Each type of smart healthcare application has different latency QA requirements. In [93], the authors proposed a computational architecture framework based on FC to reduce the delay time.

Energy Consumption: The health IoT sensors or devices in health apps use batteries, so energy-efficient solutions should be considered. Moreover, extending the lifetime of these devices also directly affects the reliability and fault tolerance of the system, especially in emergency medical scenarios [88]. To solve this problem, in [94], the authors proposed using renewable energy sources such as solar energy to power IoT devices and smart sensors.

Interoperability: The rapid growth, in both scale and number, of smart healthcare systems requires devices to interact and communicate with others in a flexible and customizable manner. This requires policymakers to quickly develop standards for connectivity, communication, and security. Currently, several standards have been proposed to standardize the healthcare domain, such as the 6LoWPAN communication protocol [95]. In our opinion, the interoperability of health devices should further focus on tackling the many threats and security vulnerabilities of the IoT era.

Privacy and Security: Another important challenge facing the IoT in general and the IoT in healthcare is security and privacy. Due to the limited resources of IoT devices, it is not feasible to implement robust security algorithms on IoT devices. In addition, health IoT devices collect large amounts of medical data that require security and privacy. If security and privacy are not ensured, the dissemination of smart healthcare systems is not feasible. Therefore, the study of lightweight security algorithms could represent a solution to these problems.

Moreover, integrating AI into the edge of networks is a possible direction for diagnosis and treatment. However, the AI training process requires powerful servers, while edge devices have limited resources; hence, approaches using lightweight AI techniques and federated learning models need to be studied. Additionally, the Metaverse and digital twins are expected to be breakthrough technologies in smart healthcare; however, due to the potential dangers to security and privacy, IoHT systems face a series of real challenges. We think that the integration of cryptographic algorithms of blockchain technology could be the key to this problem.

In our opinion, despite the dozens of challenges that still need to be addressed, smart healthcare will be an inevitable development trend in the era of the Internet of Things, and will serve to fulfil the increasing needs of people in the healthcare domain.

## 7. Conclusions

The field of IoT-based smart healthcare systems is rapidly expanding, but there are still several challenges that need to be addressed. This study presented a comprehensive survey of core technologies for smart healthcare and various computing technologies, including CC, FC, and EC. Recent studies have shown that powerful advances and integrated solutions to realizing real-time smart healthcare applications are gradually being developed. We proposed an all-in-one computing architecture framework for real-time smart healthcare applications, highlighting the advantages and challenges of the proposed architecture. A limitation of this study is that the effectiveness of the proposed computing framework is still fully evaluating yet. This aspect will be assessed by our research group in a future study. Despite these challenges, the future of medicine lies in real-time smart healthcare systems. Emerging technologies, such as the Metaverse, digital twins, and AI, will be key drivers in revolutionizing smart healthcare to serve humanity’s healthcare needs. We hope that this study will serve as an important guide and promote further research into smart healthcare in the Internet of Things era.

## Figures and Tables

**Figure 1 sensors-23-04200-f001:**
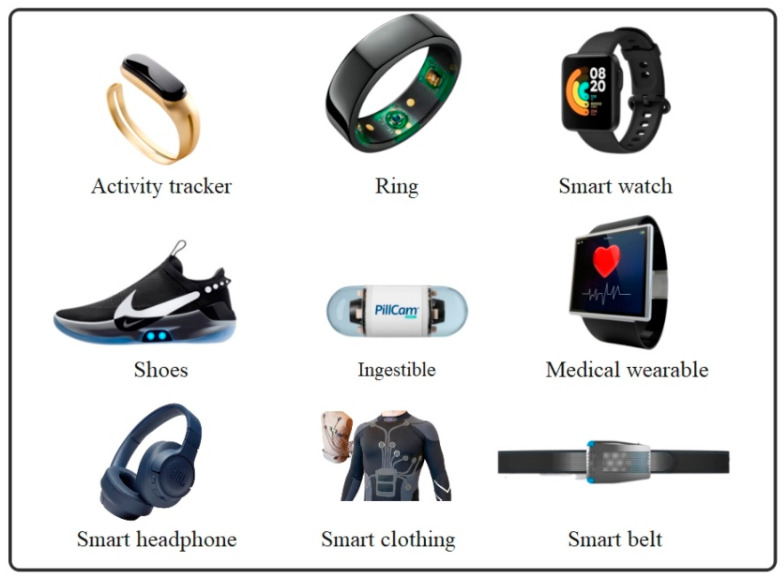
An illustration of smart wearable healthcare sensors.

**Figure 2 sensors-23-04200-f002:**
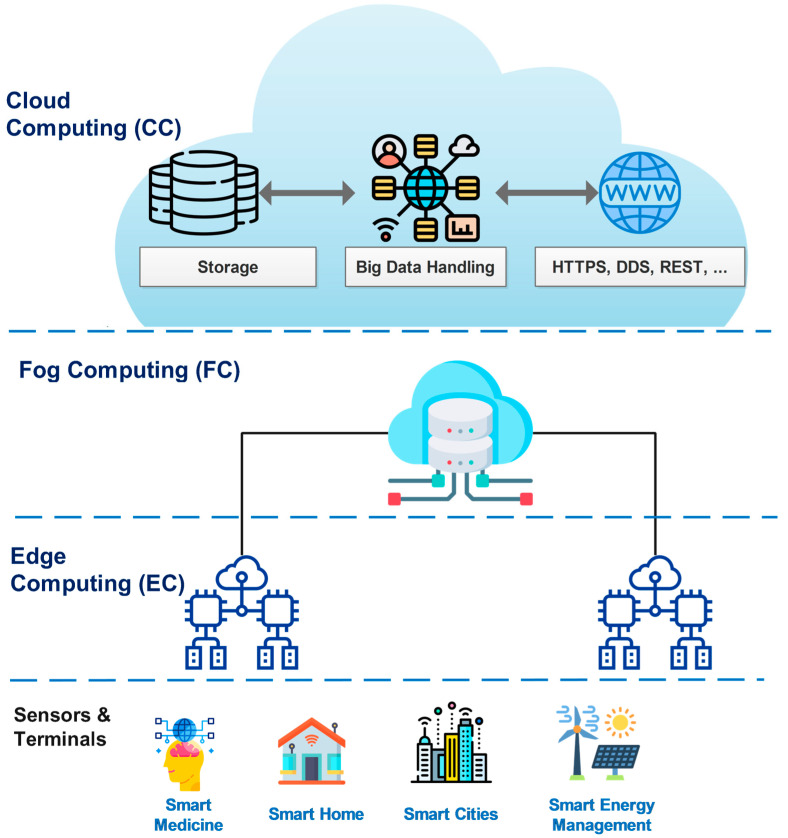
An illustration of all-in-one computing architectures for IoT ecosystems [28].

**Figure 3 sensors-23-04200-f003:**
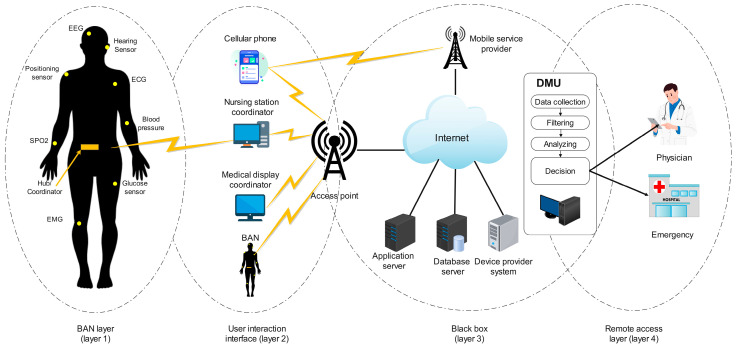
An illustration of typical WBAN architecture [30].

**Figure 4 sensors-23-04200-f004:**
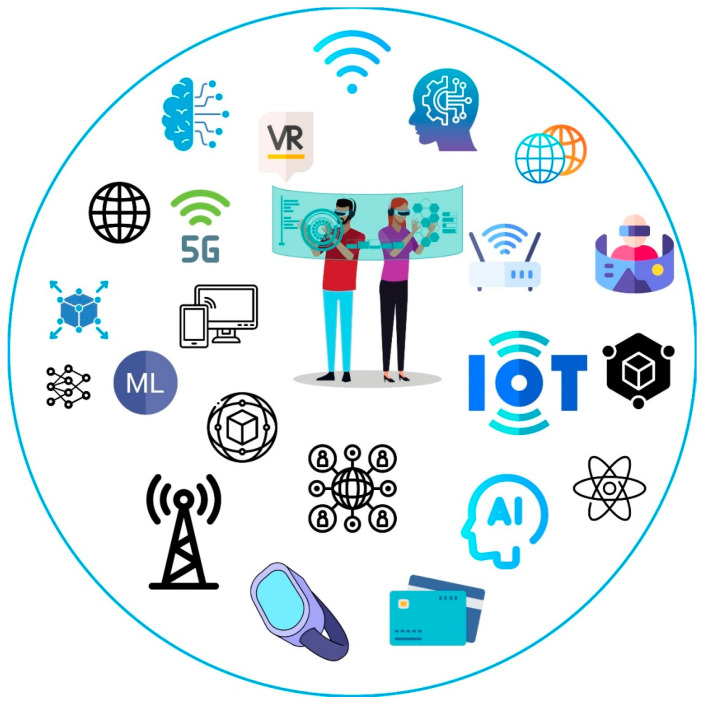
An illustration of Metaverse technology.

**Figure 5 sensors-23-04200-f005:**
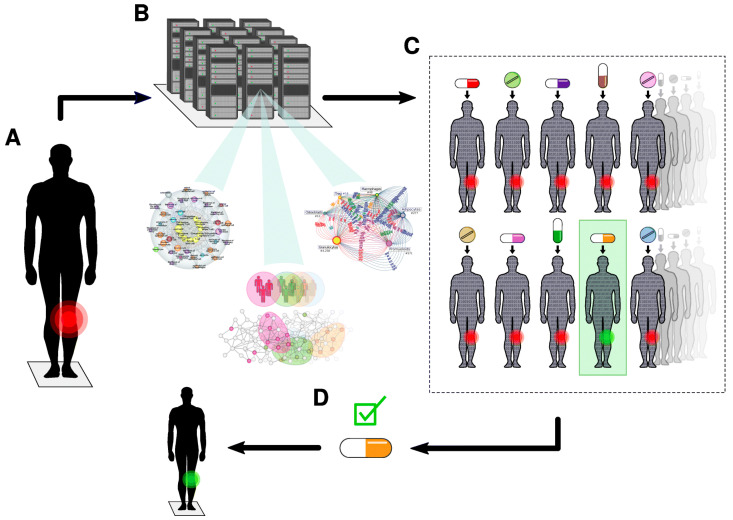
An illustration of digital twin technology.

**Figure 6 sensors-23-04200-f006:**
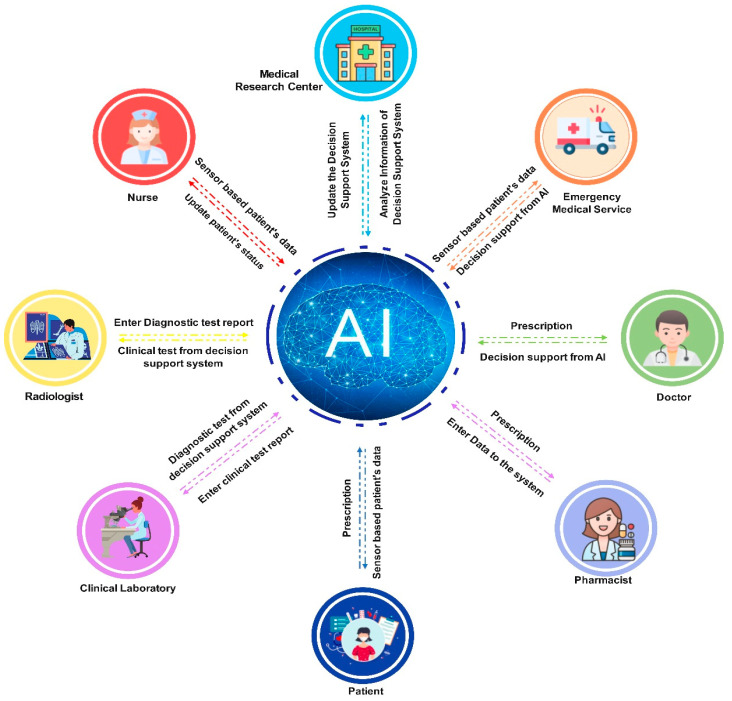
An illustration of AI-based smart health systems.

**Figure 7 sensors-23-04200-f007:**
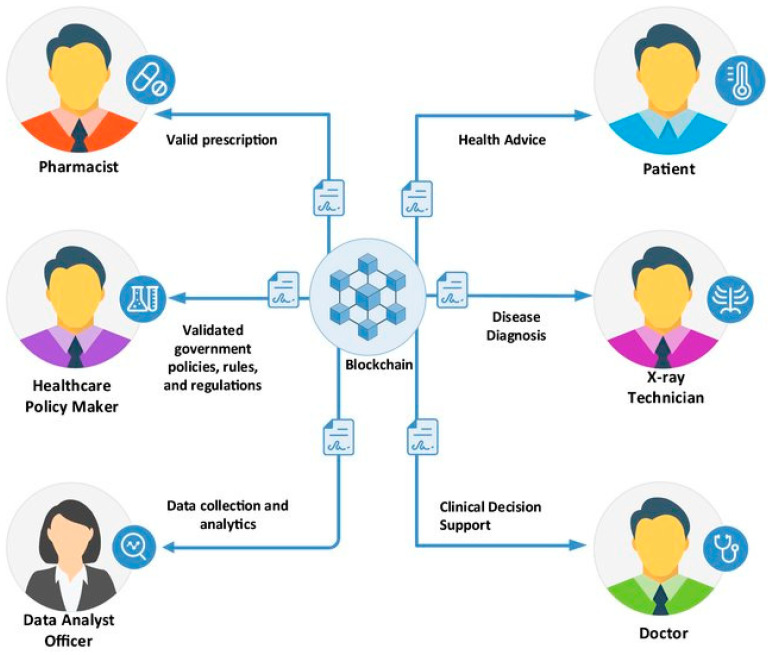
An illustration of blockchain-based smart health systems.

**Figure 8 sensors-23-04200-f008:**
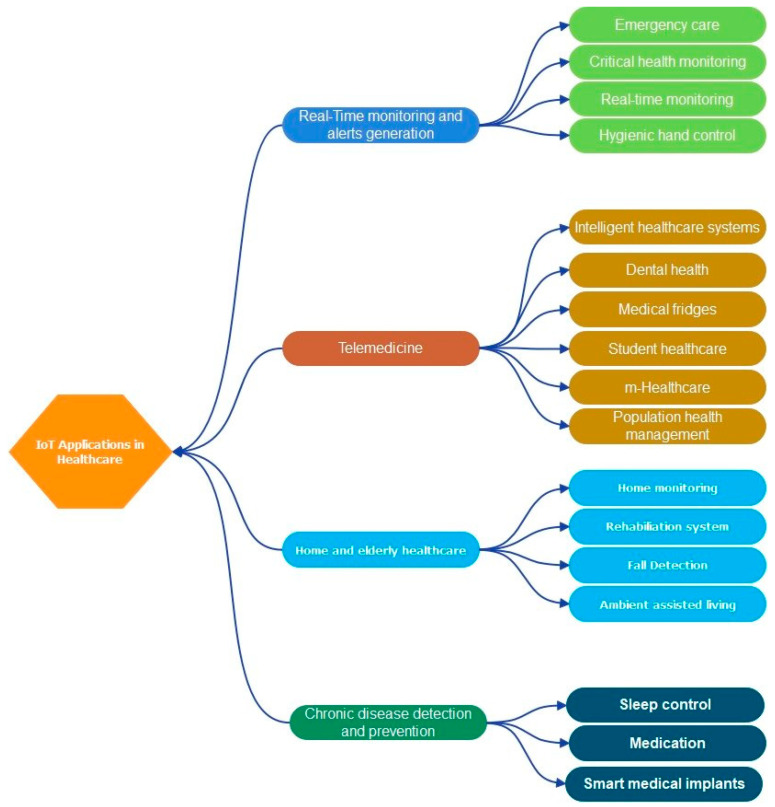
IoT-based smart healthcare systems.

**Figure 9 sensors-23-04200-f009:**
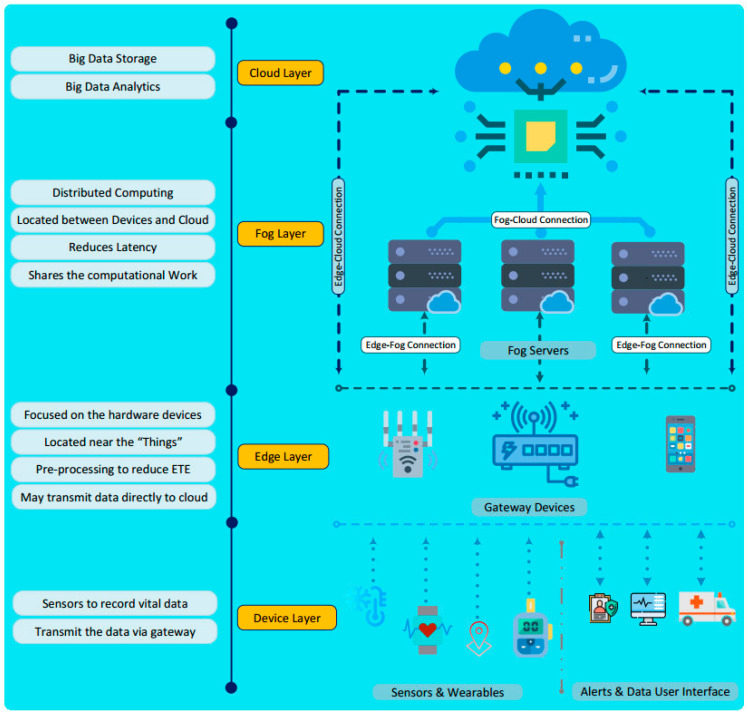
The all-in-one computing architecture for IoT-based smart healthcare applications.

**Figure 10 sensors-23-04200-f010:**
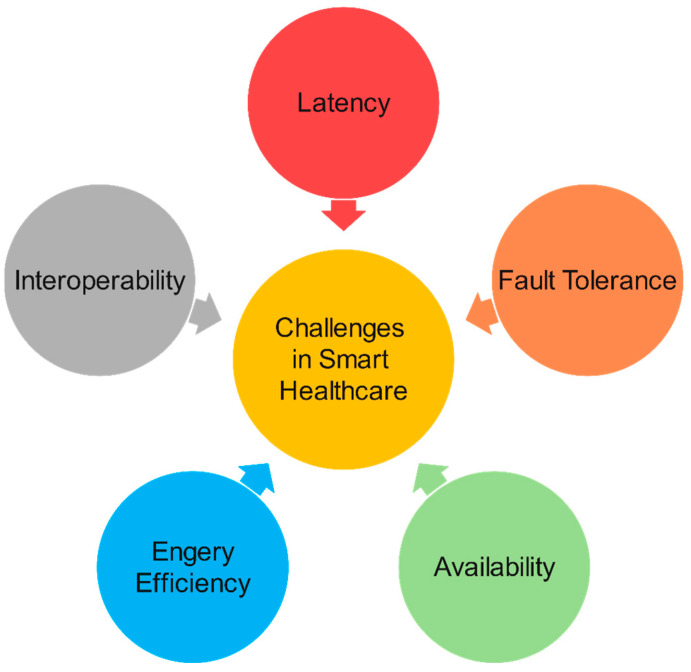
Challenges in smart healthcare.

**Table 1 sensors-23-04200-t001:** Statistics of typical research in the smart healthcare domain in the period 2019–2022.

Reference	Year	Computing	Vision	Architecture	Challenges	Technology	Results
IoT	AI	WBAN	Blockchain
Mahmoud et al. [7]	2019	CC	L	H	L	H	None	L	None	Improved energy efficiency for cloud-based IoHT applications
Habibzadeh et al. [8]	2019	CC	H	H	None	H	None	H	None	Evaluated the most advanced technologies for IoT-based clinical medical applications
Alshehri et al. [9]	2020	EC, CC	L	None	L	L	L	L	L	Reviewed IoT, AI, EC, and CC technologies, and security issues in IoHT systems.
Dian et al. [10]	2020	CC	L	None	L	H	None	M	None	Determined the integrating abilities of wearables and the IoT in IoHT systems
Qadri et al. [11]	2020	EC, FC, CC	L	H	H	H	L	M	H	Proposed some new efficient energy solutions for IoHT systems
Ullah et al. [12]	2020	CC, FC	L	H	L	H	None	L	L	Proposed efficient secure data collection, aggregation, and transmission solutions for IoHT systems
Malamas et al. [13]	2020	None	None	M	L	M	None	L	None	Proposed risk reduction methods for IoHT systems
H. Bhatia et al. [14]	2020	EC, FC	L	H	None	H	None	M	None	Proposed sensor-applied methods for IoHT systems
Amin et al. [15]	2021	EC, FC	L	H	L	H	M	L	M	Proposed an EC-based real-time IoHT systems
Jolfaei et al. [16]	2021	CC	H	M	None	M	L	M	H	Proposed blockchain-based secure IoMT systems
Dong et al. [17]	2021	FC, CC	M	M	L	H	M	M	L	Proposed an edge, fog, and cloud-based computing solution for IoHT during the COVID-19 pandemic
Taimoor et al. [18]	2021	EC, CC	L	M	M	H	H	M	M	Proposed an AI-based IoHT system
Barua et al. [19]	2021	None	None	M	H	H	None	M	M	Proposed a privacy and security solution based on the IoT and Bluetooth for IoHT systems
Aledhari et al. [20]	2022	EC, FC, CC	M	H	H	H	None	M	H	Conducted a survey of IoT contributions to healthcare and medical systems
Ali et al. [21]	2022	EC	None	H	M	M	M	None	L	Proposed an FL-based advanced security IoT system
This work	2023	EC, FC, CC	H	H	H	H	H	H	H	Fused emerging technologies into IoHT systems to enhance quality of life

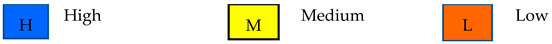

**Table 2 sensors-23-04200-t002:** Performance comparison between advanced computing technologies [28].

Characteristics	Cloud	Fog	Edge
Latency	High	Low	Low
Bandwidth	High	Low	Very low
Storage	High	Low	Low
Server Overhead	Very high	Low	Very low
Network Congestion	High	Low	Low
Energy Consumption	High	Low	Low

**Table 3 sensors-23-04200-t003:** A comparison of communication technologies for IoHT systems.

Wireless Technology	Standard	Network Topology	Transmission Range	Frequency	Bit Rate
ZigBee	802.15.4	Star, cluster tree	10–20 m	2.4 GHZ	250 Kbps
Bluetooth	802.15.1	Piconet, scatternet	10–30 m	13.56 MHz	2.1 Mbps
Low-Power Bluetooth	802.15.1	Star	~50 m	2.4–2.5 GHz	1 Mbps
IEEE 802.15.6	802.15.1	Star	<100 m	NB, UWB, HBC	75.9 Kbps −15.6 Mbps
UWB	802.15.4a	Piconet, peer-to-peer	10 m	3.1–10.6 GHz	480 Mbps
Wi-Fi	802.11	Mesh	100 m	2.4 GHz	54 Mbps
Low-Power Wi-Fi	802.11ah	One-hop	100–1000 m	780–950 MHz	150 Kbps

**Table 4 sensors-23-04200-t004:** Challenges facing Metaverse technology.

Sequence	Category	Challenges
1	Hardware Requirements	- Requirement of complex devices - Not readily available - Big size and high cost
2	Privacy and Security	- Violation of data policy - Storage of users’ information - Lack of privacy and security - Huge data processing
3	Identity Hacking	- Cloning user identifier - Account hijacking - Usage of invalid avatars
4	Neurology-Related Diseases	- Immersion in a virtual space - No real social interaction - Depression - Mental diseases
5	Digital Currencies and Payments	- Popularity of cryptocurrency types - Lack of monitoring systems and secure trading mechanisms - Flat and transparent transactions - Huge number of users
6	Law and Policies	- Virtual crime has real consequences - Conflict of international crimes - Virtual laws and policies - Policies do not keep up with the development of a virtual society

**Table 5 sensors-23-04200-t005:** Summary of typical research results of recent IoHT Applications.

Function	Research	Proposal	Core Technologies	Results
**Real-time monitoring and Alert Generation**	*Shanin* et al. [69]	IoHT system monitored ECGs, temperature, foot pressure, and heart rate	RFID, Arduino Uno, IoT	Developed a flexible, low-power electronic medical system to monitor electrocardiograms, temperature, and heart rate.
	*Swaroop* et al. [70]	IoHT system monitored temperature, and blood pressure readings and data were transmitted through different modes	IoT, Bluetooth, GSM, Wi-Fi	Developed a real-time health monitoring system design
	*Rathore* et al. [71]	The system proposal was tested using a UCI data set	Apache Spark, Hadoop Ecosystem	Proposed a scalable and real-time emergency response system
**Telemedicine**	*Zouka* et al. [72]	Neural networks and fuzzy systems used in smart healthcare	GSM, Azure IoT Hub, M2M	Proposed a smart healthcare system that provides urgent healthcare via telemedicine application and an M2M patient monitoring system
	*Rohokale* et al. [73]	The system monitored indicators such as blood pressure, sugar, and abnormal cellular growth	IoT and RFID	Proposed a healthcare monitoring system for rural people
	*Mohammed* et al. [74]	Cloud-based remote monitoring system and web services	IoT and CC	Developed an Android application called ECG Android App that provides electrocardiogram results to patients
**Home and Elderly Healthcare**	*Kumar* et al. [75]	A health monitoring system in which data are obtained by automatic neural sensors	IoT, wearable devices	Developed a health monitoring system for patients with autism
	*Onasanya* et al. [76]	Using cloud services, big data technology, and WSN in healthcare	CC, big data, WSN	Proposed IoT-based healthcare solutions for cancer patients
	*Sood* et al. [77]	Combining the IoT and fog for chikungunya epidemic detection	IoT, FC	Proposed a healthcare system to detect chikungunya and contain it at an early stage
**Chronic disease Detection and Prevention**	*Abdelgawad* et al. [78]	Using sensors to collect data and move them to the cloud for data analytics in healthcare	IoT, Bluetooth, Wi-Fi, CC	Proposed 1 IoT architecture for healthcare application
	*Yang* et al. [79]	Using the IoT to provide healthcare and real-time health monitoring.	IoT, WBAN, Zigbee, Bluetooth	Proposed a home healthcare system for wheelchair users
	*Cerina* et al. [80]	Combining the IoT cloud, ECG sensor, and GUI user interface for health monitoring	IoT and the cloud	Developed a patient health monitoring system using the IoT

**Table 6 sensors-23-04200-t006:** A comparison of the existing IoHT architectures.

Research	Year	No. of Layers	Complexity	Reliability	Real-Time Support
H. Bhatia et al. [14]	2020	5	Moderate	Moderate	Moderate
Aledhari et al. [20]	2022	3	High	Low	Moderate
Cerina et al. [81]	2017	4	Moderate	Low	Moderate
Verma et al. [82]	2018	5	High	Low	Moderate
Azimi et al. [83]	2017	3	Low	Low	Moderate
Kumar et al. [84]	2017	2	Low	Low	Low
Balakrishnan et al. [85]	2021	3	Moderate	Moderate	Moderate
Sreelakshmi et al. [86]	2021	3	Low	Moderate	Low
Mahmud et al. [87]	2018	3	Low	Moderate	Moderate
Debauche et al. [88]	2019	3	Moderate	Low	Moderate
Paul et al. [89]	2018	3	Moderate	Low	Moderate
Awaisi et al. [90]	2020	3	Moderate	Moderate	Moderate
Abdelmoneem et al. [91]	2019	4	Moderate	Moderate	Moderate

## Data Availability

Not Applicable.

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
