# Peer review of "Intelligent Healthcare: Integration of Emerging Technologies and Internet of Things for Humanity"

_sensors, 2023, doi:10.3390/s23094200_

Round 1
Reviewer 1 Report
Please define your review procedure – which scientific databases did you search, inclusion and exclusion criteria for chosen studies etc.
Figure 3. – Is this figure your work, or did you use some external source for it and need to cite it?
Please mention some similar reviews, eg. Using Internet of Things to Tackle Covid-19: A Systematic Review (https://ieeexplore.ieee.org/document/9510514) and explain how your work is different than existing similar reviews
Author Response
Dear Reviewer,
We would like to thank the reviewers who have contributed many valuable and profound comments to our research team. We have received comments and have tried to correct the article as suggested by the reviewers.
In order to convenient tracking of the updated article compared to the previous version, we summarize the changes in the enclosed PDF file.

Reviewer 2 Report
Authors presented an all-in-one computing architecture for real-time IoHT applications and presented the possible solutions of the proposed architecture. Also, they discussed challenges, open issues, and future research directions.
Following comments should be addressed in the main text to imporve quality of the paper:
1- This content presents review of Intelligent Healthcare applications in IoT. So, authors need to add review or survey word in the title, abstract and main text.
2- The categorization of existing published papers is unclear. author need to explain what is main idea to categorizae existing punlished papers and time durations.
3- Open issues and main challenges can be approved to add other hot topics.
4- Please update the references according to journal style.
Author Response

(The authors gave the same response as above.)

Reviewer 3 Report
Summary/Contributions: The authors of this report conducted a detailed survey of IoT-based medical technology and solutions. Furthermore, they offered an all-in-one computer architecture for real-time IoHT applications and presented the proposed architecture's probable solutions. Comments/Suggestions: 1. The paper is well-written and covers an interesting topic. 2. The authors are invited to add a short paragraph at the end of the introduction that describes the structure of the paper. 3. The authors may add a section that reports on similar surveys which focused on the same topic in order to emphasize the originality of their work. 4. In addition, the authors are invited to add a section about he search strategy they adopted (key words, inclusion/exclusion criteria, databases, etc.) 5. Please explain the difference between fog and edge computing. 6. More details about security aspects need to be considered. Blockchain is not the only way to secure healthcare systems. 7. The authors are invited to add a paragraph about the use of smartphone sensors for detecting specific types of diseases and monitoring the state of patients. 8. For this purpose, the authors may include the following interesting references (and others): a. https://ieeexplore.ieee.org/document/9327468 b. https://www.mdpi.com/1424-8220/19/9/2164 9. The authors need to identify the limitations of their work. 10. In the same direction, they need to propose more future work directions.
Author Response

(The authors gave the same response as above.)

Round 2
Reviewer 1 Report
The authors have improved the paper.
Reviewer 2 Report
Authors addressed comments in the revision.
Reviewer 3 Report
The authors considered all my comments and suggestions. Good luck.